# OpenReview forum: "Truthfulness Without Supervision: Model Evaluation Using Peer Prediction"
_ICLR.cc/2025/Conference — Submitted to ICLR 2025_

### Official Review · Reviewer_73k9 · 2024-10-19

**Soundness:** 1
**Presentation:** 3
**Contribution:** 2
**Rating:** 3
**Confidence:** 4

**Summary:**

This paper proposes a peer prediction mechanism for evaluating LLMs: A Model A's answers are scored by how much they help a "juror" J predict other model B's answers. The authors prove that assuming a joint prior over models' "real" answers (or a known distribution over participant's priors with some regularity conditions), reporting the "real" answers is a bayesian nash equilbrium. Experiments are conducted using a variety of different LLMs of different sizes, and on multiple different LLM benchmarks. LLMs misrepresenting their "real" answer are modeled by a prompt that tells models to provide convincing false answers.

**Strengths:**

- Exploring methods for using LLMs to aid in the evaluation of LLMs is a very timely topic
- The experiments cover a wide range of different tasks, as well as different models.
- Judging from my limited expertise in peer prediction, Theorem 2 could be of independent theoretical interest (if it is indeed novel as claimed in the appendix).

**Weaknesses:**

- The proposed approach seems to suffer from a lot of issues that are common for peer prediction mechanisms:
    - The assumption of shared priors does not appear to be very realistic, and I am not convinced the generalization from Theorem 2 helps much, as it appears to require full knowledge of the distribution over priors.
    - Honesty seems unlikely to be the only bayesian nash equilibrium. Unlike for other peer prediction mechanisms, collusion might not even be necessary for deviating from honesty in this case: If I understand correctly, a witness would obtain perfect reward if it encoded the defendant's answers in its own (in a way the juror is able to decode). However, an answer that encodes the correct answer can be very different from the correct answer in many cases.
- If I understand the experimental setup correctly, it appears to provide an unfair advantage of the proposed method over LLM-as-a-Judge, as despite the weak judge, the peer prediction mechanism has access to strong peer models. If such models are available, it seems misleading to have LLM-as-a-Judge use the weak judge model rather than one of the stronger peer models.
    - Some additional information on why LLM-as-a-judge was implemented the way it is (for example, no few-shot prompting) would also be helpful to better assess, whether the comparison is fair or an overly weak baseline [1] was chosen.
 - I am worried that the inverse scaling experiment might be confounded by a similar issue: The improvements could be caused by the increasing capability of the peers rather than the increasing capability of the evaluated model. An ablation in which the peer models are fixed to the juror model's size would be useful here.
- The experiments on incentive compatibility seem to only employ a single, non-adaptive deceptive "attack". This is insufficient to establish that the mechanism works in practice [2], especially for strong, potentially superhuman, models.
    - Considering different approaches to deception and analyzing them in more detail would also help to ensure that the observed inverse scaling is not just an artifact of the specific deceptive attack: One potential explantation for the inverse scaling would be that small models rarely understand the instruction to be deceptive and thus simply behave very similar to the honest model, making them impossible to detect.
- The paper at times makes very strong claims such as "the method [is] applicable to superhuman models" and "enabling reliable evaluation of superhuman models without trusted supervision." that seem too strong, even disregarding the potential issues with the experiments.

Nitpicks:
- The naming of the entities involved in the game is a bit confusing: Why is the Witness receiving the punishment rather than the defendant?
- The paragraph "Scaling Properties with Jury Population Size" seems to lack some details, such as what Regression the R^2 refers to.
- The font used for probability distributions is hard to read (D alone would be fine, but P,D and Q(?) all look pretty similar
-  It might be useful to provide a few basic details on how the juror model is implenented in the main text rather than only in the appendix.
- I would find the statement of theorem 1 a bit more clear if the last condition on the game's scores was stated in the beggining rather than as an if-clause in the very end.
- Figure 1 would be easier to read with subcaptions indicating the differences between subfigures. Also, the y-axis seems to show the negative of punishment for deception (otherwise, higher rather than lower would be better)


[1] Carlini, Nicholas, et al. "On evaluating adversarial robustness." arXiv preprint arXiv:1902.06705 (2019).Carlini, Nicholas, et al. "On evaluating adversarial robustness." arXiv preprint arXiv:1902.06705 (2019).

[2] Lipton, Zachary C., and Jacob Steinhardt. "Troubling Trends in Machine Learning Scholarship: Some ML papers suffer from flaws that could mislead the public and stymie future research." Queue 17.1 (2019): 45-77.

**Questions:**

- Do the results shown in figure 1 look qualitatively similar when only multiple choice or only open-ended tasks are considered?
- The numbers in the experiment from section 3 are difficult to interpret without more context: For many tasks, the gap between the 70B model and the 405B one seem small, and the 70B model even performs better in about 20% of tasks. How do these gaps relate to gaps in the models' performance on these tasks according to the ground truth (as well as according to model-as-a-judge)?
- Could you add an ablation using different models rather than instances of the same model as participants? That would be useful to rule out the following explanation for inverse scaling:
    - With its very limited training data (3 few-shot examples), the best the juror can do for honest witnesses is to predict that the defendant's answer is similar to the witness's. Meanwhile for dishonest models, it can only guess. Now assume that witness and defendant are either the same model, or a honest and a deceptive version of the same model. Then, it will be very easy to distinguish the honest and lying witness if the honest model's answers are deterministic, but harder if they have very high variance. The observed inverse scaling would now follow from the plausible assumption that larger models produce more similar answers with repeated sampling than smaller models do. However, this kind of inverse scaling would cease, as soon as models are large enough to be generally self-consistent.
- It looks like the surplus actually increases in $\alpha$. Why are no values of $\alpha$ larger than one included?
- What are the key elements of the proofs that distinguish them from the ones in in Schoenebeck & Yu (2023)?
- Regarding inverse scaling, it is currently difficult to read how the method scales with juror quality. Could you add a plot with the judge size rather than the evaluated model's size varying on the x-axis?
- In theorem 2, why is there a probability 1-$\delta$ if the equilibrium is supposed to be ex-ante? Does that mean that there are (rare) agents that have an incentive to deviate ex-ante?

---

> ### Author Response · Authors · 2024-11-24
> **Rebuttal [1/3]**
>
> Thank you, Reviewer 73k9, for your thoughtful review!
>
> ## Summary of Changes
>
> In our newly uploaded manuscript, most new content are concentrated in **page 14-17**. Namely,
> - Results from a range of new experiments (**Figure 5,6,7,8,9** newly added), including:
>   - Scaling experiments on fully heterogeneous participants (**Figure 5**). Results suggest that peer prediction indeed checks for truthfulness as opposed to mere similarity.
>   - Scaling experiments on populations containing half or more deceptive participants (**Figure 5,10,11,12**; the last 3 already existed in the submitted manuscript). The results also suggest that peer prediction checks for truthfulness as opposed to mere similarity.
>   - Scaling experiments with realistic, RLHF-trained deceptive model (**Figure 6**), based upon [MisleadLM](https://huggingface.co/jiaxin-wen/MisleadLM-QA) [1]. Results suggest that peer prediction continues to function in such settings akin to real-world deployment.
>   - Correlating peer prediction scores with ground truth accuracy, at a domain level (**Figure 7**). Results confirm that peer prediction rewards accurate answers.
>   - Visualizations of score distributions (**Figure 5,6**), for intuition-building purposes.
>   - Scaling properties of peer prediction with jury population size, for a wider range of aggregation exponent $\alpha$ (**Figure 8**). $\alpha=-1$ is confirmed to be a sweet spot.
>   - Scaling properties of *counterfactual* deception resistance (**Figure 9**). Results confirm that under peer prediction, honesty is a dominant strategy for participants.
> - Updated the main text according to presentation suggestions from reviewers; please see the detailed reply below for further explanations.
> - [Anonymized repository](https://anonymous.4open.science/r/Peer-Prediction-LLM-Eval-EB1E/) for reproducibility.
>
> ## Response to Questions and Suggestions
>
> > Could you add an ablation using different models rather than instances of the same model as participants? That would be useful to rule out the following explanation for inverse scaling:
>
> We agree that it's a possibility that would significantly reduce peer prediction's value.
>
> - Null Hypothesis: Peer prediction (PP henceforth) works only because honest answers are similar to each other (and likewise, dishonest answers are similar to each other), but the population contains a majority of honest participants, and PP basically evaluates an answer's similarity with the majority.
>
> We noticed this possibility after submission, and have been conducting validation experiments on it. Specifically, there are two ways to test the hypothesis above:
>
> 1. By having fully heterogeneous participants (using **all different models**), as you have suggested.
> 2. By making the population contain **an equal number of honest and deceptive participants**. (Note though this constraint is overly strong and might lead to false negatives, as it introduces an extra element of collusion)
>
> In **Figure 5**, we show experiment results testing both (1) and (2). Performance trends (such as the inverse scaling property) are consistent with those in Figure 1, thus **rejecting the null hypothesis**.
>
> Additionally, Row 2 of Figure 12 shows the detailed statistics when the population (4 participants) is exactly half-deceptive and half-honest, and the jury is SmolLM-360M (one of our smallest jurors). In this case, honest participants win out, refuting the hypothesis, which is good news. In fact, even when **75%** of the population are deceptive, honest participants still win out.
>   - Note that there is uncertainty as to the range of jurors that this result applies to. For instance, in some of the previous results (Figure 11), when participants and jurors are around the same size (which are the cases that peer prediction performs least competently in), deceptive participants may actually win when the population is half-and-half.
>
> > Some additional information on why LLM-as-a-judge was implemented the way it is (for example, no few-shot prompting) would also be helpful to better assess
>
> We have added experiments where LLM-as-a-Judge is implemented with few-shot prompting (6-shot for Qwen, 3-shot for SmolLM due to modest context window size), and results remain consistent; please see Figure 5 and 6. Examples are annotated by GPT-4o.
>
> > The improvements could be caused by the increasing capability of the peers rather than the increasing capability of the evaluated model. An ablation in which the peer models are fixed to the juror model's size would be useful here. [...] Could you add a plot with the judge size rather than the evaluated model's size varying on the x-axis?
>
> To clarify, the scaling results in Figure 1 **already fix participant model sizes** within each subplot, and only vary the jury model size. We have updated the caption to make it clear!

---

> ### Author Response · Authors · 2024-11-24
> **Rebuttal [2/3]**
>
> > Considering different approaches to deception and analyzing them in more detail would also help to ensure that the observed inverse scaling is not just an artifact of the specific deceptive attack: One potential explantation for the inverse scaling would be that small models rarely understand the instruction to be deceptive and thus simply behave very similar to the honest model, making them impossible to detect.
>
> We agree with your assessment that we need to consider diverse modes of deception, especially those that may likely arise in real-world failure cases of AI control [1,2,3]. We have thus tested peer prediction on MisleadLM [1], an LM which exhibits deceptive behaviors as a result of canonical RLHF training.
>
> The results, presented in **Figure 6**, uphold the inverse scaling property, even though both peer prediction and LLM-as-a-Judge find it harder to distinguish deceptive answers compared to the previous experiments, possibly due to increased convincingness of deceptive answers.
>
> > The assumption of shared priors does not appear to be very realistic, and I am not convinced the generalization from Theorem 2 helps much, as it appears to require full knowledge of the distribution over priors.
>
> Theorem 2 can be directly extended to the case where each participant $i$ has their own "prior over priors" $\mathcal{D}_i$. To show this fact, we need to verify that the honest strategy profile is indeed a Bayesian Nash equilibrium (BNE henceforth) under this "private $\mathcal{D}_i$" setting. To do that, observe that *for any participant $i$*, the property that "honest reporting is its *ex-ante* optimal strategy given all others do so" only depends on $i$'s personal belief $\mathcal D_i$ about others' beliefs, and not what the others really believe.
>
> It doesn't matter whether $\mathcal D_i$ is modelled as a distribution over $[0,1]^{n|\mathcal A|}$ (i.e. distribution over priors) or over $\mathcal P\left([0,1]^{n|\mathcal A|}\right)$ (i.e. distribution over distributions over priors), since the linearity of expected payoff means that BNE in the former case is preserved in the latter case, and $\mathcal P(\cdot)$ can simply be removed by linearity.
>
> Note that at this point, we are basically modeling hierarchical beliefs, which, in theory, would make the type-based formalism of [epistemic game theory](https://faculty.wcas.northwestern.edu/msi661/EpistemicGameTheory-131120.pdf) handy. However, we decided that introducing type notations would make things needlessly complicated without touching the problem's core, and so avoided hierarchical beliefs (those with >2 levels) in the theorem statement.
>
> We have made these considerations clear in Remark 1 of the revised manuscript.
>
> > Honesty seems unlikely to be the only bayesian nash equilibrium. [...] If I understand correctly, a witness would obtain perfect reward if it encoded the defendant's answers in its own.
>
> The witness does not see the defendant's answer, so the "perfect reward by encoding" claim is only true when the witness $w$ has *all* the private information that the defendant $d$ has (otherwise $w$ cannot show the jury those information in $d$'s answer that $w$ doesn't have), which is rare. And even when it does, it ($w$) could just choose to report all the information it has at face value without doing any encoding work (exactly the honest behavior we desire), and still get a perfect reward.
>
> In general, while there can indeed be other BNE, honesty is (among) the highest-paying BNE, as shown in the "achieves maximum ex-ante payoff among all equilibria for any agent" part of Theorem 1.
>
> AI systems lie usually becauses lying gets them strictly more reward (as is the case in, for example, experimental works demonstrating deception [1,2,3]). The fact that honesty is among the highest-paying BNE (by Theorem 1) ensures that no other equilibrium can reward any agent more than honesty does, thus removing the incentive to lie, even though not banning deception in the strict sense.
>
> > [...] it appears to provide an unfair advantage of the proposed method over LLM-as-a-Judge, as despite the weak judge, the peer prediction mechanism has access to strong peer models.
>
> We agree that this is an advantage, but it's a well-earned one. Our working assumption is that *smaller/weaker models are trusted, while larger/stronger models are not* - a standard assumption in scalable oversight [4,5,6]. The only feedback information LLM-as-a-Judge uses is that from the *trusted* judge (must be trusted because the judge's evaluation is used at face value without verification), while peer prediction can utilize the capabilities of *untrusted* models by having them as fellow participants - something LLM-as-a-Judge is unable to accommodate.

---

> ### Author Response · Authors · 2024-11-24
> **Rebuttal [3/3]**
>
> > claims such as "the method [is] applicable to superhuman models" and "enabling reliable evaluation of superhuman models without trusted supervision."
>
> Thank you for pointing this out! We agree in hindsight, and are tuning down the strength of these claims in the revised draft. Those expressions were originally meant to convey our motivation for carrying out the research, and we didn't intend to claim that we've fully solved these problems in any sense of the word, but we now realize our phrasing seems to have indicated otherwise. We regret our mistake, and have corrected them in the revised draft.
>
> > Nitpicks:
>
> We think these are good points - thanks! We have addressed all these issues in the revised draft, except the role naming one; we thought hard about this, but unfortunately failed to find a better set of names. Please let us know if you have suggestions about alternative naming schemes!
>
> > In theorem 2, why is there a probability 1-δ if the equilibrium is supposed to be ex-ante?
>
> Thanks for catching this! The final sentence of Theorem 2's statement should be corrected as "...is, ex ante (when the distribution $\mathcal D$ and the instantiation of all $\mathcal P^{\mathrm{J}}_j$ are known by the agents), an $\epsilon$-BNE." In other words, we consider the priors of the jurors as already instantiated, and $\delta$ is a probability about what those instantiated priors look like. Indeed, part of formula $(3)$ comes exactly from applying tail inequalities to the instantiation of the jurors' priors.
>
> > Do the results shown in figure 1 look qualitatively similar when only multiple choice or only open-ended tasks are considered?
>
> Most questions in the dataset are either multiple-choice or short-answer questions (e.g. a number theory problem with a number as the answer). We ask the participants to provide detailed reasoning along with the answer, and evaluate their reasoning along with the answer, thereby effectively using the questions as open-ended ones.
>
> In our newly added **Figure 7**, we add the *accuracy* metric, which matches the models' output choice/short answer against the ground-truth choice/short answer (ignoring the reasoning), resulting in a 0/1 score for each question. We use Gemma2-27B to perform this identity checking (since robustly extracting the choice/short answer from the output and then checking semantic identity is non-trivial), and spot-checked its judgments to ensure reliability.
>
> Correlation plots in **Figure 7** suggests that peer prediction score is a reliable proxy of ground-truth accuracy on our dataset.
>
> > For many tasks, the gap between the 70B model and the 405B one seem small, and the 70B model even performs better in about 20% of tasks. How do these gaps relate to gaps in the models' performance on these tasks according to the ground truth (as well as according to model-as-a-judge)?
>
> **Figure 7(b)** finds consistently positive correlations between peer prediction score gaps and ground-truth accuracy gaps, for all pairs of models.
>
> > It looks like the surplus actually increases in α. Why are no values of α larger than one included?
>
> We had hoped to include a wider range of $\alpha$, but did not have the time to do so. We have now plotted those other values in **Figure 9**, with $\alpha=-1$ remaining the best-performing exponent - turns out it's a unimodal rather than monotone relationship.
>
> > What are the key elements of the proofs that distinguish them from the ones in in Schoenebeck & Yu (2023)?
>
> For theorem 1: As mentioned in the draft right before the proof, the general idea is the same. The key difference is in extending Schoenebeck & Yu (2023)'s proof from their 3-agent setting to our $n$-agent setting, still using their techniques.
>
> For theorem 2: The proof of theorem 2 is quite different, and we don't think there is an analog/counterpart in Schoenebeck & Yu (2023). One could intuitively think of it as theorem 1 + generalization bound (in the statistical learning theory sense), where each agent optimizes against a finite sample of fellow agents drawn from $\mathcal D$, and we need to show that optimization against this sample doesn't deviate too far away from optimization against $\mathcal D$  itself. The general direction of Theorem 2's proof is thus similar in spirit to proofs of statistical generalization bounds, but using quite different techniques (U-statistics, instead of VC dimension or Rademacher complexity).
>
> We have added these explanations into our revised draft as Remark 2.
>
> ---
>
> Thank you again for the extremely helpful review. Please let us know what you think!

---

> > ### Author Response · Authors · 2024-11-24
> > **References**
> >
> > [1] Wen, Jiaxin, et al. "Language models learn to mislead humans via rlhf." *arXiv preprint arXiv:2409.12822 (2024).*
> >
> > [2] Lang, Leon, et al. "When Your AIs Deceive You: Challenges of Partial Observability in Reinforcement Learning from Human Feedback." *arXiv preprint arXiv:2402.17747 (2024).*
> >
> > [3] Williams, Marcus, et al. "Targeted Manipulation and Deception Emerge when Optimizing LLMs for User Feedback." *arXiv preprint arXiv:2411.02306 (2024).*
> >
> > [4] Burns, Collin, et al. "Weak-to-strong generalization: Eliciting strong capabilities with weak supervision." *arXiv preprint arXiv:2312.09390 (2023).*
> >
> > [5] Leike, Jan, et al. "Scalable agent alignment via reward modeling: a research direction." *arXiv preprint arXiv:1811.07871 (2018).*
> >
> > [6] Khan, Akbir, et al. "Debating with more persuasive llms leads to more truthful answers." *arXiv preprint arXiv:2402.06782 (2024).*

---

> > > ### Comment · Reviewer_73k9 · 2024-11-24
> > >
> > > Thank you for the detailed response.
> > >
> > > As the discussion period ends in a few days and thorougly evaluating the new experiments likely requires another detailed read of the paper, it is likely that I won't be able to reply in more detail before the end of the discussion period. I do however promise to go through the newly presented evidence and potentially update my score, before the AC decisions.
> > >
> > > Some immediate questions from a shallow reading of the response:
> > >
> > > > To clarify, the scaling results in Figure 1 already fix participant model sizes within each subplot, and only vary the jury model size
> > >
> > > Is the jury model the same as the juror model?
> > >
> > > > The witness does not see the defendant's answer, so the "perfect reward by encoding" claim is only true when the witness has all the private information that the defendant has.
> > >
> > > If I understand correctly, the same statement would be correct in a slightly modified form, replacing "perfect" by optimal (given the available information) and "encoding the answer" by "providing maximal information about the answer in a way only the juror can understand". Is that correct?
> > >
> > > > The only feedback information LLM-as-a-Judge uses is that from the trusted judge (must be trusted because the judge's evaluation is used at face value without verification), while peer prediction can utilize the capabilities of untrusted models by having them as fellow participants - something LLM-as-a-Judge is unable to accommodate.
> > >
> > > Just to clarify, this statement is derived from the game-theoretic analysis? If in practice someone gave me an LLM to use a a juror/defendant without a way to verify that it follows the incentives of the game you defined, there is no guarantee that the method would actually work, right?
> > >
> > > > Theorem 2 can be directly extended to the case where each participant has their own "prior over priors"
> > >
> > > If that is indeed correct, I would suggest replacing theorem 2 by a statement like this, as it seems like a substantially more realistic setting.

---

> > > > ### Comment · Reviewer_73k9 · 2024-11-24
> > > >
> > > > Another clarification question:
> > > >
> > > > Do you think of your proposed method as a method for general model evaluation, or as a method for detecting deception more specifically? Some of the framing seems to point at the former, while most experiments are focused on the latter.

---

> ### Author Response · Authors · 2024-11-26
> **Response to 11.24 Followup**
>
> Thank you for the prompt response!
>
> Since ~all extra content is concentrated in **page 14-17** (1 page of text + 3 pages of figures), please feel free to only read those new pages, without re-reading the rest of the paper.
>
> Feel free to follow up further! The discussion deadline has been extended to 2nd, though the PDF revision deadline remains 27th.
>
> > Is the jury model the same as the juror model?
>
> Yes, that's a typo. Nice catch!
>
> > If I understand correctly, the same statement would be correct in a slightly modified form
>
> Exactly, but the second half of our response was indeed aimed at such a modified critique. To quote our response:
>
> - *In general, while there can indeed be other BNE, honesty is (among) the highest-paying BNE, as shown in the "achieves maximum ex-ante payoff among all equilibria for any agent" part of Theorem 1.*
> - *AI systems lie usually becauses lying gets them strictly more reward (as is the case in, for example, experimental works demonstrating deception [1,2,3]). The fact that honesty is among the highest-paying BNE (by Theorem 1) ensures that no other equilibrium can reward any agent more than honesty does, thus removing the incentive to lie, even though not banning deception in the strict sense.*
>
> Here we are saying: regardless of how profitable encoding one's private information is (it need not be perfect reward!), there's always the option of "not encoding it and share the private info as is" that's at least as profitable.
>
> This is true in general, not only for encoding-style deception, but also for all other non-honest equilibria. This should remove the incentive for deception.
>
>
> > If in practice someone gave me an LLM to use a a juror/defendant without a way to verify that it follows the incentives of the game you defined, there is no guarantee that the method would actually work, right?
>
> Great question. Theorem 1,2 don't provide theoretical guarantee in that case, but we do have experimental evidence (Figure 5/6/12 where ≥half of the participants are deceptive) that peer prediction usually functions well as a measure of truthfulness **even outside the honest equilibrium** defined in Theorem 1/2. In other words, peer prediction can tolerate a decent amount of deception by multiple agents, while still maintaining discriminative power wrt deception.
>
> To solve the root of this lack-of-incentive issue, one may also train/finetune models using peer prediction scores as reward, essentially performing unsupervised training for truthfulness. This would make the peer prediction score the explicit optimization objective of the LM policy, assuming no goal misgeneralization ([Langosco et al.](https://arxiv.org/abs/2105.14111)).
>
> If this works, such an unsupervised truthfulness training process could use training data beyond the volume/difficulty limits of human annotation. It could be implemented in the model alignment pipeline. *Whether it indeed works is a hypothesis to be tested by training experiments, not a foregone conclusion.*
>
> Given the existing volume/workload of this paper, and the volume/workload/downstream analysis required by the training experiments, we think they would merit a followup paper. (Exploring a new method outside the Overton window is a lot of work, yet dividing the work into paper-sized chunks might mean facing critiques to each chunk due to the information left in the other chunks, which is a bit of a difficult situation.)
>
> > replacing theorem 2 by a statement like this, as it seems like a substantially more realistic setting
>
> Makes sense! We've now explicitly included in Theorem 2's statement that "The same is true when agents hold disagreeing 'prior over priors' $\mathcal{D}_i$," and laid out the reduction (from the disagreeing $\mathcal{D}_i$ case to the shared $\mathcal{D}$ case) in the appendix.
>
> We intentionally avoided a strictly formal definition of what "hold disagreeing priors over priors" means, and instead went for an intuitive treatment (which should be clear on its own!), since these are 3rd-order belief hierarchies, and a strict formalism could be confusing to read and distracts from the core idea.
>
> > a method for general model evaluation, or as a method for detecting deception more specifically?
>
> The most accurate description would be *peer prediction evaluates **truthfulness***, i.e. the extent to which the evaluated answer matches the truth.
>
> A distinction famously exists between truthfulness and honesty (cf. Diagram 2 in [Evans et al.](https://arxiv.org/pdf/2110.06674)): honesty requires saying things the model itself believes to be true (even if the belief could be mistaken), while truthfulness requires saying things that are actually true. Truthfulness asks the model to be not only *honest* but also *informed*.
>
> As a result, we conducted both model honesty experiments (e.g. Figure 1,5) and model informativeness/correctness experiments (e.g. Figure 3,7) when testing the validity of peer prediction as an evaluation metric of truthfulness.

---

> > ### Comment · Reviewer_73k9 · 2024-11-28
> >
> > Thank you for the response.
> >
> > > The most accurate description would be peer prediction evaluates truthfulness, i.e. the extent to which the evaluated answer matches the truth.
> >
> > I find this statement confusing. It seems to me like the method can clearly not reliably evaluate truthfulness, unless the other models involved in the evaluation know the truth?

---

> ### Author Response · Authors · 2024-11-28
>
> > It seems to me like the method can clearly not reliably evaluate truthfulness, unless the other models involved in the evaluation know the truth?
>
> The peer prediction metric, as a proxy measure for truthfulness, operates by assuming that (a majority of) other participants know the truth. (Theorem 1 in particular assumes everyone is Bayesian and thus can't be confidently wrong; Theorem 2 relaxes this to some extent by allowing prior disagreements.)
>
> In practice though, truths tend to be more consistent than falsehoods (barring collusion), which give them asymmetric advantage over falsehoods. For instance, a correct mathematical calculation can be helpful for predicting another correct mathematical calculation, as the two share the same set of rules; on the other hand, a mistaken mathematical calculation is less helpful for predicting another mistaken mathematical calculation, as mistakes are made in random ways.
>
> Indeed, we see that in practice, peer prediction can measure truthfulness even when falsehood-tellers constitute ≥half the population (cf. Figure 5/6/12). Though of course, this doesn't always work, and still breaks down when falsehoods are sufficiently prevalent and the model capability gap is at the lower end of the inverse scaling property (cf. Figure 11).

---

> > ### Comment · Reviewer_73k9 · 2024-11-28
> >
> > Dear authors,
> >
> > After spending some additional time reading the rebuttal and the paper, I still cannot help but find the inverse scaling results extremely implausible. If I understand correctly, they seem to indicate that the proposed method seems to *consistently perform better when worse jury models are used*. I find it difficult to believe that this result would generalize to realistic settings and also fail to see, how the result would be explained by the theory.
> >
> > This makes me very reluctant to raise my score, despite the evident effort put into the rebuttal.
> >
> > Could you provide any intuition/explanation, on why you believe the inverse scaling results are more than an artifact of the specific evaluation method, and would generalize?
> >
> > I also had a look into the posted repository to get a better understanding, but found things difficult to parse. Would it maybe be possible for you to share a csv file that contains the participants response and juror scores for your main experiment on MMLU (or a representative subset, in case that is too much data) in an easily accesible way?

---

> ### Author Response · Authors · 2024-11-29
> **Response to 11.28 Followup**
>
> Dear Reviewer 73k9,
>
> Thanks for the response, and thank you for your evaluation efforts! We have shared a lot of our thoughts, and we appreciate that you have sticked with us.
>
> Here's a [new anonymized repository](https://anonymous.4open.science/r/Peer-Prediction-Data/README.md) containing the requested full data files. The questions are randomly shuffled, so feel free to take any sample from it. Please let us know if there are other files you'd like to see, or if you have further questions!
>
> > Could you provide any intuition/explanation, on why you believe the inverse scaling results are more than an artifact of the specific evaluation method, and would generalize?
>
> - When the jury and the defendant have very similar capability levels, the jury can mostly predict the defendant's answer without much help from the informant, as it already has most of the objective-level knowledge that the defendant and the informant have.
> - When the defendant is stronger than the jury, the jury can no longer single-handedly predict the defendant's answer, and rely more on the informant. The honest informant thus gets more reward by helping the jury make better predictions; the deceptive informant in turn gets lower rewards, as it now harms the jury's prediction quality to a larger extent.
>
> **Intuitively**, Alice teaching a 10 year old to solve an AMC math question is stronger indicator of Alice's expertise than if she taught a high school student to solve the same question, as the high schooler may be able to do 80% of the work even without Alice's help.
>
> **To compare the intuition with our theory**: in Theorem 1/2, we are using private information to model capability gaps, which is accepted practice in theoretical modeling of alignment [[1](https://arxiv.org/abs/2402.17747)]. Such modeling is most accurate when capability gaps are large - when Bob is immensely outsmarted by Alice, it makes Alice's thoughts literally beyond Bob's reach, making those thoughts effectively private information. This may mean that in practice, Theorem 1/2 applies most squarely in cases where capability gaps are large; and these are exactly the cases where peer prediction perform the best. Since they are also the worst-case/highest-priority scenarios for safety and alignment, we think this would not harm the practical applicability of peer prediction.
>
> ---
>
> If we erase the jury's prior knowledge about the question at hand, making it entirely dependent on the informant, we predict that the discriminative power of peer prediction will be enhanced in cases where model capability gaps are smaller (jury and participants being closer in capabilities), seemingly reducing/reverting the inverse scaling trend. If this is true, peer prediction's performance will become uniformly better than the results we presented, which should be good news. However, we suspect that doing such erasing is largely equivalent to reducing model size (and thus reducing capabilities in general) - exactly the x-axis of the inverse scaling trend. Thus, even if it's true, it wouldn't really challenge the inverse scaling property.
>
> Finally, we agree that a never-ending inverse scaling property is too good to be true. At some point, e.g. when the jury become weak enough that it can't even reason properly, we expect the trend to be reversed, where larger capability gap means lower discriminative power. However, we have used the smallest modern LLM we can find (SmolLM-135M) as jury, and at least until SmolLM-135M, the inverse scaling trend continues.
>
> We were also initially surprised to discover the inverse scaling property, and have experimented with changing the mechanism (e.g. disallowing the jury to see past informant answers during in-context learning, to prevent the jury from identifying and adapting to deceptive informants), in order to rule out setting specificity. However, results stay broadly consistent in those experiments, which convinced us of the explanation given above. We don't currently have those experiment results in an organized/presentable manner, but could organize/redo them if needed.

---

> > ### Comment · Reviewer_73k9 · 2024-11-30
> >
> > Thank you for the response.
> >
> > I tried to open one of the gz files on two different operating systems but got an error message.
> >
> > This explanation looks somewhat similar to some of the concerns I voiced in my initial review regarding the jury model essentially "repeating" the witness answer. But in that case, it seems as if the impact of a dishonest defendant would conceptually be very similar to the impact of a dishonest judge in the llm-as-judge setting (with the main difference that a) dishonest predictions and dishonest judgments can look somewhat different and b) multiple judges are used in the peer prediction setup.). This reinforces my worry about the fair comparison to llm-as-judge in terms of model access.
> >
> > After all, while I appreciate the authors' engagment during the rebuttal, I have decided to keep my score.
> >
> > For future versions, I strongly recommend to de-emphasize or further qualify the inverse scaling results, as I do think that they are misleading with respect to the practical scalability of the method (for which scaling in the participants' size would be more relevant). In addition, I do recommend using a more fair comparison to LLM as a judge, where both methods have access to the same other models.
> >
> > As a side note, after going through the figures several times now: The x-axis on the figures is very confusing. Rather than showing the size gap (which leaves open which of the two models is scaled), it would be way more clear if it just stated "jury model size".

---

> ### Author Response · Authors · 2024-12-01
> **Thank You**
>
> Thank you for your response and your evaluation efforts! We respect that you've made your decision, and decided to voice our disagreements nonetheless, if only for reference. Please feel free to ignore these remarks.
>
> - On the concern around repeating the witness answer: **(1)** Peer prediction empirically remains truthful when $\geq$half answers are deceptive, but LLM-as-a-Judge could not remain truthful when $\geq$half judgments are deceptive. **(2)** Conceptually, it's because the jury *uses its reasoning capabilities to exploit the **logical** cross-consistency that's distinct to truthful answers* (cf. the "mistaken mathematical calculation" example in our discussion thread). This fundamentally sets the jury apart from a mere repeater, and it's exactly this difference that led to the distinctive empirical property in (1).
>
> - On broken files: We looked into the issue, and it seems like Anonymous Github somehow also scanned our compressed files for de-anonymizing strings, which broke the files. We have uploaded the files to [Github](https://github.com/AnonZQ5M3VY9/Peer-Pred-Data) instead. Please feel free to ignore them though.
>
> Finally, we'd like to express our appreciation for the highly helpful comments you've made throughout this discussion.

---

> > ### Comment · Reviewer_73k9 · 2024-12-02
> >
> > > Peer prediction empirically remains truthful when half answers are deceptive, but LLM-as-a-Judge could not remain truthful when half judgments are deceptive.
> >
> > While I see the intuition behind this, this claim seems quite dependent on implementation details. In particular, it seems likely to me that your observations would not hold up with deceptive participants that are optimized to break the mechanism. Also, I would not be surprised if you'd observe similar results with LLM-as-juge, depending on the specific implementation. Further experiments substantiating this could strenghten future versions of the paper.

---

> ### Author Response · Authors · 2024-12-04
> **Final Remarks**
>
> We really appreciate that you've got back to us!
>
> > I would not be surprised if you'd observe similar results with LLM-as-juge, depending on the specific implementation
>
> To the best of our knowledge, there has been no attempt in the literature [1] to extend LLM-as-a-Judge to a multiple-judge setting, and it remains an open question how such a method is to be implemented. Therefore, while comparisons of peer prediction with this extended LLM-as-a-Judge could strengthen our position, we find it unreasonable to require that such comparisons against *not-yet-proposed* methods be made.
>
> Among all the possible population-based evaluation methods, peer prediction is, to our knowledge, the only one with solid theoretical support, and thus we chose peer prediction as our focus (while comparing it with baselines already present in the literature).
>
> > This explanation looks somewhat similar to some of the concerns I voiced in my initial review regarding the jury model essentially "repeating" the witness answer.
>
> As can be seen in the data files that we share, informant and defendant answers are almost always substantially and semantically different, so it seems unclear to us how the "repeating" explanation could be applied.
>
> One could argue that the jury may simply be evaluating semantic similarity between answers, but we believe this explanation has already been ruled out by experiment results featured in our *Rebuttal [1/3]* (please search for "Null Hypothesis").
>
> ---
>
> Thank you again for your engagement, and for the feedback you've provided! We appreciate this exchange.
>
>
>
> [1] Gu, Jiawei, et al. "A Survey on LLM-as-a-Judge." arXiv preprint arXiv:2411.15594 (2024).

---

### Official Review · Reviewer_RBqK · 2024-11-04

**Soundness:** 2
**Presentation:** 3
**Contribution:** 3
**Rating:** 6
**Confidence:** 3

**Summary:**

The authors propose "peer prediction" as a novel method to evaluate LLMs without requiring trusted supervision or ground truth labels. The method works by measuring how well one model's answers help predict another model's answers through a jury system, with theoretical guarantees that honest and informative answers are optimal. Through theoretical analysis and experiments the authors demonstrate three key findings: (1) the method effectively distinguishes model capabilities across diverse domains, (2) it exhibits an inverse scaling property where resistance to deception actually increases as the capability gap between jury and participants grows larger, enabling reliable evaluation of superhuman models, and (3) the method's resistance to deception improves with larger participant and jury populations.

**Strengths:**

- The idea of exploring mechanisms that exhibit game-theoretic incentive compatibility in model evaluation is pretty interesting and sufficiently new.
- The problem is important and well-motivated.
- Section 3 is well-written and from what I was able to check technically correct.

**Weaknesses:**

I don't think the paper has any major technical issue, but it could be improved in terms of clarity for people not familiar with the mechanism design literature. Maybe add a background section in the appendix. The figures can also be improved by making the font larger and writing subfigure captions. Finally, I would also like to point that being more technical when defining terms like "being more truthful", "superhuman", is extremely helpful. I was able to understand the paper regardless and I understand the use of these non-technical terms has increased in this literature, but I recommend to be more precise in the final version of the work if possible.

**Questions:**

- Can you explain the practical implications of assumption 1? I think the paper lacks a discussion about practical implication of all the assumptions in Sec 3.
- I had a hard time understanding Fig 3, can you expand on the details of each subfigure?

---

> ### Author Response · Authors · 2024-11-25
> **Rebuttal [1/2]**
>
> Thank you, Reviewer RBqK, for your helpful feedback!
>
> ## Summary of Changes
>
> In our newly uploaded manuscript, most new content are concentrated in **page 14-17**. Namely,
> - Results from a range of new experiments (**Figure 5,6,7,8,9** newly added), including:
>   - Scaling experiments on fully heterogeneous participants (**Figure 5**). Results suggest that peer prediction indeed checks for truthfulness as opposed to mere similarity.
>   - Scaling experiments on populations containing half or more deceptive participants (**Figure 5,10,11,12**; the last 3 already existed in the submitted manuscript). The results also suggest that peer prediction checks for truthfulness as opposed to mere similarity.
>   - Scaling experiments with realistic, RLHF-trained deceptive model (**Figure 6**), based upon [MisleadLM](https://huggingface.co/jiaxin-wen/MisleadLM-QA) [1]. Results suggest that peer prediction continues to function in such settings akin to real-world deployment.
>   - Correlating peer prediction scores with ground truth accuracy, at a domain level (**Figure 7**). Results confirm that peer prediction rewards accurate answers.
>   - Visualizations of score distributions (**Figure 5,6**), for intuition-building purposes.
>   - Scaling properties of peer prediction with jury population size, for a wider range of aggregation exponent $\alpha$ (**Figure 8**). $\alpha=-1$ is confirmed to be a sweet spot.
>   - Scaling properties of *counterfactual* deception resistance (**Figure 9**). Results confirm that under peer prediction, honesty is a dominant strategy for participants.
> - Updated the main text according to presentation suggestions from reviewers; please see the detailed reply below for further explanations.
> - [Anonymized repository](https://anonymous.4open.science/r/Peer-Prediction-LLM-Eval-EB1E/) for reproducibility.
>
> ## Response to Questions and Suggestions
>
> > I had a hard time understanding Fig 3, can you expand on the details of each subfigure?
>
> Certainly.
>
> Take the top-left subfigure as example: the x-axis is a collection of problem domains (public relations, marketing, ...), and for each of these domains we have 3 ranges plotted - red, blue, and green. The red range represents the mean score that the 8B model received from peer prediction (Algorithm 1) when answering test questions from this domain; blue for 70B; and green for 405B.
>
> The top-left subfigure contains 10 domains, and the remaining 7 subfigures each similarly contains 10-11 domains, so that's a total of 85. These 85 domains together include 37079 test questions.
>
> It can be seen that 405B generally outperforms 70B, and 70B generally outperforms 8B, which is expected.
>
> To further demonstrate the ability of peer prediction scores to tell better answers from worse ones, we also found consistently positive correlation between peer prediction scores (which, recall, is calculated without ground truth labels) and ground-truth accuracy (**Figure 7**).

---

> ### Author Response · Authors · 2024-11-25
> **Rebuttal [2/2]**
>
> > Can you explain the practical implications of assumption 1? I think the paper lacks a discussion about practical implication of all the assumptions in Sec 3.
>
> That's a good point! We have included the discussion below as Remark 3, which we now refer to when presenting Assumption 1.
>
> Let's first examine the first part of Assumption 1, **(bounded) variability within prior** (VWP henceforth), which asks that PMI between different participants is bounded.
>
> *Pointwise mutual information (PMI) is a measure of how much the actual probability of a particular co-occurrence of events p(x, y) differs from what we would expect it to be on the basis of \[sheer coincidence\].* [2] *\[It\] draws on the intuition that the best way to weigh the association between two words is to ask how much more the two words co-occur in a corpus than we would have expected them to appear by chance.* [3]
>
> Here, PMI is instead taken over participants' answers - VWP is measuring the association between different participants, asking "when Alice and Bob both answers D to the question, how much we expect that to be because they converge upon the truth, compared to sheer coincidence?"
>
> The second part, **(bounded) variability across priors** (VAP henceforth), asks that when two agents with disagreeing priors assign differing prior probabilities to "another participant (e.g. Alice) giving a certain answer (e.g. D)", the ratio between their probabilities is bounded.
>
> **Taken together**, there are usually two ways is which Assumption 1 is satisfied in the real world. Both are sufficient conditions, so we **only need one to be true**.
>
> 1. **Lower-bounded probabilities** (VWP+VAP). In a 4-option multiple-choice question, likely everyone always assign no less than 0.5% probability to any option, just in case they are wrong. In this case, we can verify that VWP and VAP always hold.
> 2. **All participants have uncertainties about the answer** (VWP) and **participants all know that others have uncertainty too** (VAP). In this case, VWP is satisfied because when Alice and Bob both answers D, the "sheer coinincidence" explanation can now no longer be ruled out, given that both Alice and Bob's response has some randomness in it. VAP is satisfied because, if both you and I agree that Alice has some "stable" uncertainty between options A/B/C/D, we won't disagree catastrophically (e.g. by >1000x) on how likely it is for Alice to answer D.
>
> Note that these aren't necessary conditions, but rather two most plausible reasons for VWP/VAP being true in the real world; there are likely more of them.
>
> > it could be improved in terms of clarity for people not familiar with the mechanism design literature.
>
> Good point. We have thought about having an independent background section or expanding the related works section to introduce the mechanism design formalism.
>
> However, the field of algorithmic mechanism design has a very diverse range of formalisms (for example, the formalism of peer prediction is entirely different from that of auction design) [4], and so we decided to cover only the formalism of peer prediction to avoid distracting the reader. We did this introduction in Section 3. Please let us know if there are things in this introduction that we could a better job explaining!
>
> ---
>
> Thank you again for this helpful review! We would love to hear what you think, and would love to know if you have had updates upon reading our response or the new suite of experiment results.
>
> ## References
>
> [1] Wen, Jiaxin, et al. "Language models learn to mislead humans via rlhf." *arXiv preprint arXiv:2409.12822 (2024).*
>
> [2] Bouma, Gerlof. "Normalized (pointwise) mutual information in collocation extraction." *Proceedings of GSCL 30 (2009): 31-40.*
>
> [3] Jurafsky, Daniel. *"Speech and language processing." (2000).*
>
> [4] Nisan, Noam, and Amir Ronen. "Algorithmic mechanism design." *Proceedings of the thirty-first annual ACM symposium on Theory of computing (1999).*

---

### Official Review · Reviewer_6HAK · 2024-11-04

**Soundness:** 2
**Presentation:** 2
**Contribution:** 3
**Rating:** 5
**Confidence:** 3

**Summary:**

When discussing evaluation methods for language models, the strong reliance on supervision and the unavailability of reliable supervision on hard tasks, particularly in scalable oversight, lead to the exploration of “evaluation methods without reliable supervision”. Therefore, this paper proposes the “peer prediction method”, which leverages “game-theoretic incentive compatibility” from mechanism design literature, to perform resistance to deception without trusted supervision.

This paper first clarifies the importance of exploring evaluation methods, explains their inspiration for leveraging “game-theoretic incentive compatibility”, and then highlights the merits of their “peer prediction method”. The “peer prediction method” applies several models as “participants” and some separate agents as “jurors”, then evaluates participants’ answers to held-out questions by assessing their ability to help jurors predict others' responses, using peer answers as targets instead of ground-truth labels. In the experiment section, the paper applied a dataset containing questions spanning across tremendous domains to test the effectiveness and resistance to deception, including a finding of “Inverse Scaling Properties” and other ablation studies on scaling properties.

**Strengths:**

This paper proposes a novel method that does not need reliable supervision. It is resistant to deception and has a strong scaling performance.

This paper priorly takes “game-theoretic incentive compatibility” into consideration and provides mathematical proof for their theorem

**Weaknesses:**

1. The scenario of exploring the topic of “evaluate models without supervision” is not well defined. According to this paper, “lack of reliable supervision” occurs in “scalable oversight”, which is well explained. Therefore, it is reasonable to discuss this scenario but only limited to it. For other scenarios, “aiming to ensure that they are safe, reliable, and beneficial” (Line 57) does not necessarily lead to the motivation to “evaluate models without supervision”. The claim of “trusted supervision for difficult tasks is often unavailable” (Line 12) in the abstract does make sense but lacks sufficient and concrete examples (only “scalable oversight” is mentioned). Further elaboration and explanation of “under what circumstances when trusted supervision is unavailable” should be provided.

2. Some of the conclusions are based on strong assumptions. Take Line 270 to Line 277 as an example, it is reasonable to have the conclusion of “peer prediction method is incentive compatible”, but the conclusion of “In particular, models are incentivised to converge upon honest and informative policies, if either (I) they are trained on the peer prediction scores as reward signals, or (II) they perform inference-time reasoning to maximize the evaluation scores” is likely to lead based on a strong assumption that efficient benign answers are required. If tremendous malicious answers are proposed, according to the “incentive capability”, the models may be incentivized to converge upon deceptive results.

3. The methodology strongly relies on the combination of several LMs. Considering the peer prediction method takes peer answers as targets instead of ground-truth labels, the results of the models are interactional. That means if one of the models is changed, the performance of other models will comparatively be changed. The provided experiments lack examples of different combinations of LMs as participants.

4. This method will be really resource-consuming when considering superhuman models. Plus,” distinguish better models from worse ones” is not a good evaluation metric to me as it is a relative result instead of an absolute one, which means it will have more limitations. For example, you need to find appropriate models for comparison when testing

**Questions:**

1. Could you please provide further explanation for what is mentioned in weakness 1 and weakness 2?

2. According to weakness 3, are you available to provide more experiments considering different combinations of the participant's model?

---

> ### Author Response · Authors · 2024-11-24
> **Rebuttal [1/2]**
>
> Hi Reviewer 6HAK - thank you for your feedback! We've found your comments to be highly instructive.
>
> ## Summary of Changes
>
> In our newly uploaded manuscript, most new content are concentrated in **page 14-17**. Namely,
> - Results from a range of new experiments (**Figure 5,6,7,8,9** newly added), including:
>   - Scaling experiments on fully heterogeneous participants (**Figure 5**). Results suggest that peer prediction indeed checks for truthfulness as opposed to mere similarity.
>   - Scaling experiments on populations containing half or more deceptive participants (**Figure 5,10,11,12**; the last 3 already existed in the submitted manuscript). The results also suggest that peer prediction checks for truthfulness as opposed to mere similarity.
>   - Scaling experiments with realistic, RLHF-trained deceptive model (**Figure 6**), based upon [MisleadLM](https://huggingface.co/jiaxin-wen/MisleadLM-QA) [1]. Results suggest that peer prediction continues to function in such settings akin to real-world deployment.
>   - Correlating peer prediction scores with ground truth accuracy, at a domain level (**Figure 7**). Results confirm that peer prediction rewards accurate answers.
>   - Visualizations of score distributions (**Figure 5,6**), for intuition-building purposes.
>   - Scaling properties of peer prediction with jury population size, for a wider range of aggregation exponent $\alpha$ (**Figure 8**). $\alpha=-1$ is confirmed to be a sweet spot.
>   - Scaling properties of *counterfactual* deception resistance (**Figure 9**). Results confirm that under peer prediction, honesty is a dominant strategy for participants.
> - Updated the main text as per presentation suggestions from reviewers; please see the detailed reply below for further explanations.
> - [Anonymized repository](https://anonymous.4open.science/r/Peer-Prediction-LLM-Eval-EB1E/) for reproducibility.
>
> ## Response to Questions and Suggestions
>
> > “aiming to ensure that they are safe, reliable, and beneficial” (Line 57) does not necessarily lead to the motivation to “evaluate models without supervision”.
>
> It's a great question - thanks for raising it! Here is our motivation.
>
> Before we deploy strong (and possibly superhuman) models into the real world, we need to evaluate their safety and trustworthiness [1,2] to prevent harms, including, importantly, harms from deception and manipulation [3]. Indeed, such evaluation has been the official policy of [leading](https://www.anthropic.com/news/anthropics-responsible-scaling-policy) [AI](https://cdn.openai.com/openai-preparedness-framework-beta.pdf) [labs](https://deepmind.google/discover/blog/introducing-the-frontier-safety-framework/).
>
> However, when the model that we evaluate is itself pushing the frontier of AI capabilities (e.g. when OpenAI released GPT-4), there is, by definition, no stronger model to supervise it. Human evaluators are very often also deceived or misled by convincing models [3,4,5,6], and thus cannot be fully trusted to provide supervision. This fact that no trustworthy supervision exists when evaluating frontier models, is what motivates our work on supervision-free evaluation.
>
> > Further elaboration and explanation of “under what circumstances when trusted supervision is unavailable” should be provided.
>
> As argued in the paragraph above, trusted supervision is unavailable when **the model being evaluated is itself pushing the frontier of AI capabilities**. This is because the model is stronger than any other model that could supervise it (we demonstrated in Figure 1 that evaluating strong models with a weak LLM-as-a-Judge is prone to deception), and the model is often capable of exploiting human supervision as well [3,4,5,6].
>
> > “In particular, models are incentivised to converge upon honest [...]” is likely to lead based on a strong assumption that efficient benign answers are required. If tremendous malicious answers are proposed, according to the “incentive capability”, the models may be incentivized to converge upon deceptive results.
>
> Indeed, incentive compatibility (Theorem 1 and Theorem 2) shows that *when other agents are honest*, the model is incentivized to be honest as well. It does not rule out the possibility of deception when other agents are deceptive, but *neither does it imply that deception is incentivized in such cases* - intuitively speaking, there is an asymmetry between the payoffs of honest and deceptive strategies when information-theoretic mechanisms (like peer prediction) are used [7], with honesty generally having an advantage. Underlying this intuition is the *Data Processing Inequality* in Appendix C.1.
>
> Ultimately, it all comes down to experiments. Our experiment results show that when $50\\%$ (**Figure 5,12**) or sometimes even $75\\%$ (**Figure 12**) of the population are deceptive, honest strategies can still win against deceptive strategies. This indicates that peer prediction is effective in incentivizing honesty, even in the presence of deception by others.

---

> ### Author Response · Authors · 2024-11-24
> **Rebuttal [2/2]**
>
> > According to weakness 3, are you available to provide more experiments considering different combinations of the participant's model?
>
> Certainly. In addition to the model combinations we already presented (`Llama3.1-8B/70B/405B` + `Mistral-7B-v0.3` for informativeness, [`Llama3.1-8B` or `Gemma-2-2B/27B`]+ [`Qwen2.5-0.5B/1.5B/3B/7B` or `SmolLM-135M/360M`] for honesty), we have added experiments with fully heterogeneous participants in **Figure 5** (`Llama3.1-8B`, `Gemma-2-9B`, `Mistral-7B-v0.3`) and **Figure 6** (`Llama2-7B`, `Gemma-2-9B`, `Mistral-7B-v0.3`), and also populations with varying honest-deceptive compositions ($25\\%$, $50\\%$, $75\\%$) in **Figure 12**. These experiments show that peer prediction is overall effective in incentivizing honesty across a wide range of participant combinations.
>
> > This method will be really resource-consuming when considering superhuman models.
>
> In fact, for models that push the frontier of AI capabilities, we believe that peer prediction induces **lower** cost than alternative methods for evaluation.
>
> Currently, the only widely accepted "gold standard" for evaluating AI models is human judgment [8,9], which is expensive and time-consuming (not to mention its susceptibility to deception [3,4,5,6]).
>
> As for AI-based evaluation methods such as LLM-as-a-Judge, they are typically only used for *evaluating weaker models using stronger models* (e.g. using GPT-4o to evaluate outputs of a finetuned Llama 8B), not the other way around (which is widely considered unreliable). As a result, evaluation of newly released frontier models (whose capabilities surpass existing models) is still done using human judgment or human-written datasets [9], which is expensive and time-consuming.
>
> Peer prediction, on the other hand, is a scalable and deception-resistant evaluation method that can be used to evaluate frontier models, while being much cheaper and faster than human evaluation.
>
> > “distinguish better models from worse ones” is not a good evaluation metric to me as it is a relative result instead of an absolute one, which means it will have more limitations. For example, you need to find appropriate models for comparison when testing
>
> We believe that a relativist evaluation metric is no less useful than an absolutist one, and indeed, the former can be easily converted to the latter by using, for example, the Elo score system.
>
> Take **LMSYS Chatbot Arena** [9] for example, where models are paired up and compared against each other in a tournament-style competition, using human evaluation. While each round of comparison is relativist, Chatbot Arena's Elo score system converts these relative comparisons into an absolute ranking of models, where each model is assigned a numerical score that reflects its overall performance. Today, Chatbot Arena is the *de facto* gold standard for benchmarking chatbots.
>
> We believe that a similar system can be implemented for peer prediction, where models are paired up and compared against each other in a tournament-style competition, and the results are converted into an absolute ranking of models using the Elo score system. This approach would share the same benefits as Chatbot Arena, but at a tiny fraction of the cost and time required, since peer prediction, unlikely Chatbot Arena, requires no human evaluation.
>
> We consider the construction of such evaluation infrastructure subject of future work, while our current paper aims to lay down the methodological foundations.
>
> ---
>
> Thank you again for the highly insightful feedback! Please let us know what you think, and we encourage you to reevaluate our work in light of the new experiments and explanations we have provided.
>
> ## References
>
> [1] Bengio, Yoshua, et al. "Managing extreme AI risks amid rapid progress." *Science 384.6698 (2024): 842-845.*
>
> [2] Shevlane, Toby, et al. "Model evaluation for extreme risks." *arXiv preprint arXiv:2305.15324 (2023).*
>
> [3] Park, Peter S., et al. "AI deception: A survey of examples, risks, and potential solutions." *Patterns 5.5 (2024).*
>
> [4] Wen, Jiaxin, et al. "Language models learn to mislead humans via rlhf." *arXiv preprint arXiv:2409.12822 (2024).*
>
> [5] Lang, Leon, et al. "When Your AIs Deceive You: Challenges of Partial Observability in Reinforcement Learning from Human Feedback." *arXiv preprint arXiv:2402.17747 (2024).*
>
> [6] Williams, Marcus, et al. "Targeted Manipulation and Deception Emerge when Optimizing LLMs for User Feedback." *arXiv preprint arXiv:2411.02306 (2024).*
>
> [7] Kong, Yuqing, and Grant Schoenebeck. "An information theoretic framework for designing information elicitation mechanisms that reward truth-telling." *ACM Transactions on Economics and Computation (TEAC) 7.1 (2019): 1-33.*
>
> [8] Chen, Guiming Hardy, et al. "Humans or llms as the judge? a study on judgement biases." *EMNLP 2024.*
>
> [9] Chiang, Wei-Lin, et al. "Chatbot arena: An open platform for evaluating llms by human preference." *arXiv preprint arXiv:2403.04132 (2024).*

---

> ### Author Response · Authors · 2024-12-01
> **We Agree (Response to Nov 29 Followup)**
>
> Thank you for your followup response!
>
> > Would you mind adding what you respond to “‘distinguish better models from worse ones’ is not a good evaluation metric…” on your conclusion-future direction part?
>
> Yes, we agree this is important, and commit to including that explanation in our camera-ready version.
>
> The rebuttal-stage deadline for uploading revised manuscript has unfortunately passed (it seems that the response wasn't visible to us until the 29th), and the next time we are able to do this will be in the camera-ready version. Given that this comment is fully public, we are making this a credible commitment and will make sure to follow through.
>
> > It is fair to try some less powerful models to propose and verify your method, but you have to make it clear and well-explained in your paper
>
> This makes sense. In our current manuscript on OpenReview, we have already removed the word "superhuman" from all substantial claims we make about peer prediction, and instead only used it when explaining the background and motivation. (For instance, it now occurs only once in the abstract, when introducing the motivation.)
>
> In addition to this, as per your suggestions, we commit to doing the following in our camera ready version:
> - Include the following sentence in the abstract: *"While peer prediction hasn't been tested on strictly superhuman models, its ability to reliably supervise models much stronger than the jury hints at its potential applicability to superhuman models."*
> - Add a paragraph of text in introduction, next to the "our contributions" paragraphs, restating and explaining the *"While peer prediction hasn't been tested on strictly superhuman models [...]"* sentence above.
> - In section 3, after line 240, add a paragraph explaining our setting, namely (1) that we are using "evaluating stronger models with much weaker juries" as a proxy for our "evaluating frontier-pushing models" motivation, and (2) that we are motivated by frontier-pushing models due to the lack of reliable means to evaluate them.
>
> Finally, a quick explanation on why we did the experiments we did:
> - During our study, Llama3.1-405B came out, with [benchmark results](https://ai.meta.com/blog/meta-llama-3-1/) comparable to GPT-4o and Claude 3.5 Sonnet at the time. We thus considered it to be at least *at* the frontier of capabilities (if not actively pushing it), and included it into our experiments. We acknowledge though, it's not necessarily still at the current frontier now, and due to budget constraints we only tested 405B in our informativeness experiments.
> - Since the frontier is always shifting, we think the unchanging core of our setting is "evaluating a model without using any other supervisor with comparable capabilities", rather than whether any model is at the frontier at any specific time. However, we do strongly agree we need to make this setting very clear, and we aim to do so in the changes we committed to above.
>
> > I would also suggest reorganizing the content related to Theorem 1, mainly the paragraph [...]
>
> We agree. We commit to adding the following paragraph right after Theorem 1, before line 270:
>
> *While Theorem 1 focuses on an all-honest equilibrium, our later experiments show that when $50\\%$ (Figure 5,12) or sometimes even $75\\%$ (Figure 12) of the participants are deceptive, honest strategies are still favored over deceptive ones. This indicates that, in practice, peer prediction is effective in incentivizing honesty, even in the presence of deception by others.*
>
> ---
>
> Thank you again for the very helpful comments! They have been highly instrumental to improving our manuscript. We could also share our draft of the additional paragraphs we committed to adding, if you wish to see them, and if that's allowed by ICLR rules.
>
> Please feel free to let us know of any additional questions.

---

### Meta-Review · Area_Chair_ayJe · 2024-12-23

**Metareview:**

## Summary:
This paper addresses the challenge of evaluating language models without reliable supervision, especially for complex tasks where trusted supervision is lacking. It introduces a peer prediction method inspired by game-theoretic incentive compatibility from mechanism design literature. This method distinguishes between honest and informative answers and deceptive ones without relying on ground truth labels. The paper demonstrates the effectiveness and resistance to deception of peer prediction through theoretical guarantees and empirical validation on models up to 405B parameters. Compared to the LLM-as-a-Judge approach requiring strong and trusted judges, peer prediction exhibits an inverse scaling property where resistance to deception strengthens as the gap between the jury and participants widens. This enables reliable evaluation of powerful models without trusted supervision, showcasing the potential for game-theoretic resistance to model deception in alignment and evaluations.

## Strengths:
1. The paper introduces a novel method of evaluating LLMs using other LLMs without requiring reliable supervision or ground truths. Compared to LLM-as-a-judge approaches, it highlights the resistance to deception and the inverse scaling property (weaker/smaller jury + stronger/larger participants improves the evaluation).
1. The paper incorporates "game-theoretic incentive compatibility" principles, which are interesting to the community and well-motivated. They also provided mathematical proofs for the theorems presented. Theorem 2 can be of independent interest.
1. The experiments are comprehensive, covering various tasks and models.
1. The paper is well written and clear in presentation.

## Weaknesses:
1. The observed inverse scaling property might be a trivial consequence of better answers from the larger honest witnesses and that the smaller jury model simply follows the answers without much extra reasoning or judgment. Moreover, it is not intuitive to get improved evaluations from worse jury models without a convincing analysis. And it is not clear when the property no longer hold if we keep decreasing the jury size. Although the intuitions presented in the discussion are helpful, they do not provide sufficient information to exclude the possibility.
1. The paper overclaims the importance of their contribution to evaluating superhuman models since the experiments are conducted on non-superhuman open-source models. The authors revised the statements accordingly later by removing "superhuman models" but this also weakens the motivation of "no trusted supervision for superhuman models".
1. The motivation from "the lack of trusted supervision" might not hold for many practical tasks and applications where the ground truths or rewards are not very expensive to attain. This may limit the scope of this paper.
1. There is a gap between the theory and the empirical results: the theory assumes that the majority of participants know the truths, while the empirical results show a better robustness to >50% deceptive participants. This also may imply that the majority vote might be a much simpler but hard baseline to beat (even when >50% are deceptive peers as long as their answers are diverse but honest peers' answers are consistent).
1. The scalability and computational cost of the proposed method can be much worse than the LLM/human judge approaches as it requires inferences on multiple models.


## Decision:
The authors provided detailed clarifications and additional experimental results in the rebuttal, as requested by the reviewers. Two out of the three reviewers responded to the rebuttal and actively participated in multiple rounds of in-depth discussions with the authors. The remaining reviewer who has not responded voted for borderline acceptance with relatively low confidence. Although some important concerns have been addressed by the rebuttal and discussion, the two reviewers in the discussion still decided to keep their original ratings since several significant issues have not been comprehensively investigated as summarized in the above weakness session. The meta-reviewer carefully read all the discussions and the paper as well. Despite the importance of the studied peer evaluation, the key assumptions and problem setups in this paper are not fully convincing given the current theoretical and empirical results. The authors are encouraged to revise the draft according to the discussion before submitting it to the next conference.

**Additional Comments On Reviewer Discussion:**

The authors provided detailed clarifications and additional experimental results in the rebuttal, as requested by the reviewers. Two out of the three reviewers responded to the rebuttal and actively participated in multiple rounds of in-depth discussions with the authors. The remaining reviewer who has not responded voted for borderline acceptance with relatively low confidence. Although some important concerns have been addressed by the rebuttal and discussion, the two reviewers in the discussion still decided to keep their original ratings since several significant issues have not been comprehensively investigated as summarized in the above weakness session. The meta-reviewer carefully read all the discussions and the paper as well. Despite the importance of the studied peer evaluation, the key assumptions and problem setups in this paper are not fully convincing given the current theoretical and empirical results. The authors are encouraged to revise the draft according to the discussion before submitting it to the next conference.

---

### Decision · Program_Chairs · 2025-01-22

Reject